# Spatial Manipulation of Particles and Cells at Micro- and Nanoscale via Magnetic Forces

**DOI:** 10.3390/cells11060950

**Published:** 2022-03-10

**Authors:** Larissa V. Panina, Anastasiya Gurevich, Anna Beklemisheva, Alexander Omelyanchik, Kateryna Levada, Valeria Rodionova

**Affiliations:** 1Institute of Novel Materials and Nanotechnology, National University of Science and Technology MISiS, 119049 Moscow, Russia; gurevich-anastasia@yandex.ru (A.G.); annabekl@yandex.ru (A.B.); 2REC “Smart Materials and Biomedical Applications”, Immanuel Kant Baltic Federal University, 236041 Kaliningrad, Russia; 9azazel@gmail.com (A.O.); kateryna.levada@gmail.com (K.L.); valeriarodionova@gmail.com (V.R.)

**Keywords:** magnetic force, gradient magnetic field, diamagnetic levitation, micromagnetic array, cell arraying

## Abstract

The importance of magnetic micro- and nanoparticles for applications in biomedical technology is widely recognised. Many of these applications, including tissue engineering, cell sorting, biosensors, drug delivery, and lab-on-chip devices, require remote manipulation of magnetic objects. High-gradient magnetic fields generated by micromagnets in the range of 10^3^–10^5^ T/m are sufficient for magnetic forces to overcome other forces caused by viscosity, gravity, and thermal fluctuations. In this paper, various magnetic systems capable of generating magnetic fields with required spatial gradients are analysed. Starting from simple systems of individual magnets and methods of field computation, more advanced magnetic microarrays obtained by lithography patterning of permanent magnets are introduced. More flexible field configurations can be formed with the use of soft magnetic materials magnetised by an external field, which allows control over both temporal and spatial field distributions. As an example, soft magnetic microwires are considered. A very attractive method of field generation is utilising tuneable domain configurations. In this review, we discuss the force requirements and constraints for different areas of application, emphasising the current challenges and how to overcome them.

## 1. Introduction

Magnetic fields can freely penetrate biological tissues and, in combination with magnetic micro- and nanoparticles, are widely exploited for diagnostic and therapeutic applications. Numerous reviews are devoted to magnetic manipulation, since this area of research is very versatile, and problems are approached from different angles [1]. Typical examples include the transport of objects for targeted drug delivery, hyperthermia and other localised therapies [2,3,4], particle and cell trapping, and colloidal assembly for tissue engineering and cell arraying [5,6]. Various magnet systems are used for specific applications, and the purpose of this review is to discuss the basic concepts involved in their design.

Magnetic particles respond differently to homogeneous and spatially distributed magnetic fields. Strong homogeneous magnetic fields polarise the particles, which can be used to increase their clustering via dipole interactions. An example of this application is the promotion of stem cell aggregation. Here, mesenchymal stem cells labelled with magnetic nanoparticles were patterned with the use of an external magnetic field to produce continuous and large cartilage tissue substitutes [7]. Magnetic nanoparticles functionalised with antibodies for death receptors were able to trigger apoptosis due to the clustering effect [8]. Large uniform magnetic fields can affect the diffusion of charged particles such as Na^+^, K^+^, and Ca^2+^ ions and plasma proteins through the Lorentz force [9,10,11]. It has also recently been shown that large uniform magnetic fields can stimulate the diffusion of paramagnetic and diamagnetic species inside cells [12]. Living cells are not in thermodynamic equilibrium, and many species, such as paramagnetic free radicals and molecules, can have significant concentration gradients [13]. The gradient-concentration force is small, and its role becomes important only in the presence of ultrahigh magnetic fields.

Magnetic fields with large spatial gradients impose a sufficient force on magnetic particles (paramagnetic or diamagnetic) and are even capable of altering cell functions [14,15,16]. The magnetic properties of many biological species are weak, although they can be enhanced by proper contrast [17,18] or by magnetic particle labelling and internalisation [19,20,21,22,23,24,25]. Nevertheless, weak magnetic properties require the development of magnetic systems that generate both relatively high field strength and high gradients. This is possible with strong permanent magnets having micron-sized poles. For example, a magnetic system produced by sputtering NdFeB onto a micro-patterned silicon wafer satisfies both conditions [26,27]. Other examples include the use of ferromagnetic microwires [28,29].

Magnetic manipulation strategies are continuously developing. The main concern is related to short-range forces since a magnetic field decreases rapidly with the distance from magnetic poles. Another problem is related to the particle magnetic moment and the magnetic field occurring in the same direction, which causes a paramagnetic particle to move in the direction of the highest field gradient (or the lowest for a diamagnetic particle). Therefore, magnetic systems should be specifically designed to overcome some of these problems. To controllably direct magnetic particles, a combination of a strong uniform magnetic field and a weaker spatial field was proposed [30,31]. A system with oppositely polarised magnets creates a field configuration with a zero-field point in the centre and a nearly linear field gradient away from this point. This configuration is of interest for magnetic particle imaging [32,33] and for increasing the penetration of particles, even into solid tumours [34].

High-gradient magnetic fields generated by magnets of various shapes have been utilised in regenerative medicine to grow cellular structures and organs in vitro. The tissue structure was constructed from layers of co-cultured hepatocytes and endothelial cells [35]. The spatial placement of cells and the ability to overcome the intercellular interaction were realised through the use of the Mag-TE system, consisting of Fe_3_O_4_ nanoparticles with a liposome coating and a system of cubic and spherical neodymium magnets. Urothelial and vascular systems of several layers of human aortic endothelial cells, human aortic smooth muscle cells, and mouse NIH 3T3 fibroblasts were created using a cylindrical neodymium magnet [36].

Magnetic field gradients can be exploited for controllable and scalable assembly of para- and diamagnetic objects into two-dimensional (2D) structures [37,38]. The possibility of constructing human umbilical vein endothelial cells with different linear structures with a thickness of about one cell using the field of a permanent magnet and thin steel plates for cell micropatterning was shown [39].

Multiparametric characterisation of individual cells also requires reproducible planar arrays of cells. These cell grids allow monitoring the behaviour of a small number of cells with specific properties (e.g., rare tumour cells) in a large population. For these investigations, cell arraying techniques in suspension are of particular interest.

This paper is organised as follows. We start by analysing the magnetic forces that are imposed on magnetic particles. In Section 3, a summary of the computation of magnetic fields produced by uniformly magnetised cylinders is presented as a basis for designing a complex magnetic system. Section 4 analyses the main magnetic systems from single magnets to arrays of magnets for manipulating magnetic particles, including diamagnetic trapping. Section 6 considers the effect of a gradient magnetic field on particle diffusion in various systems, including magnetic grids and ferrite garnet films with a tunable domain structure. Finally, Section 7, the problem of biocompatibility between cell cultures and glass-coated ferromagnetic wires is briefly discussed.

## 2. Gradient Magnetic Field and Forces

For electrically neutral paramagnetic and diamagnetic objects characterised by a magnetic moment p, the dipole force $F_{m}$ exerted on them in a magnetic field H is defined by the gradient of potential energy $U_{m}$:(1)$$F_{m}=-\nabla U_{m},\;\;U_{m}=-\left(p\cdot B\right),\;B=\mu_{0}H$$
where B is the magnetic induction and $\mu_{0}$ is the permeability of the vacuum. For magnetically weak objects, p is linearly proportional to the external field and, for particles of ellipsoidal shapes, is expressed as
(2)$$p_{i}=V_{p}(\chi_{p}-\chi_{ex})\frac{\left(1+\chi_{ex}\right)H_{i}}{\left(1-N_{i}\right)\chi_{ex}+N_{i}\chi_{p}+1}$$

In Equation (2), $p_{i}\;$and $H_{i}$ are the components of the particle magnetic moment and external field along the principal axes of the ellipsoid, $N_{i}$ is the corresponding demagnetisation factor, $V_{p}$ is the particle volume, and $\chi_{p}$ and $\chi_{ex}$ are the susceptibilities of the particle and the environment (dispersion solution, including contrast agents). For spherical particles, $N_{i}=1/3$, and Equations (1) and (2) are simplified to
(3)$$p=4\pi b^{3}(\chi_{p}-\chi_{ex})\frac{\left(1+\chi_{ex}\right)H\left(r\right)}{2\chi_{ex}+\chi_{p}+3}$$
(4)$$F_{m}=4\pi b^{3}\mu_{0}(\chi_{p}-\chi_{ex})\frac{\left(1+\chi_{ex}\right)}{2\chi_{ex}+\chi_{p}+3}\;\nabla {\left(H\left(r\right)\right)}^{2}$$
where b is the particle radius. In the case of a cluster of particles, when the demagnetising factor is not known, the particle polarisability $\chi_{ap}$ can be introduced, which is found experimentally by measuring the magnetisation curve M(H):(5)$$p=V_{p}\chi_{ap}H,\;\;\chi_{ap}=\frac{dM}{dH}$$

Depending on the sign of $\left(\chi_{p}-\chi_{ex}\right)$, the magneto dipole force defined by Equation (4) will either attract particles to regions with higher magnetic fields (relatively paramagnetic objects) or push them into regions with a minimal field (relatively diamagnetic objects). This means that the paramagnetic particles move towards the sources of the external magnetic field, where the field is at its maximum. In the case of diamagnetic objects, there is an attractive variant of their capture in the regions of local energy minima that can form between the magnetic poles. In order to compete with other forces, such as elastic and viscous forces, the magnetic dipole force must be in the range of 10–50 pN [40,41]. For example, for a microsized particle with a susceptibility of 2 (as for cells with internalised iron oxide nanoparticles), a magnetic field gradient of 10 kT/m is sufficient to reach the value of $F_{m}≅30\;pN$. The diamagnetic susceptibility of cells and other biological objects is small, so much larger gradients are required. In this case, the magnetic effect can be enhanced by using paramagnetic contrast, as follows from Equation (2). For this purpose, solutions of paramagnetic salts (e.g., Ho(NO_3_)_3_) or Gd can be used [18].

The magnetic field of the desired configuration can be generated by combining permanent magnet films with soft magnetic grids [38,39]. The magnetic field is almost uniform at the surface of permanent magnet films, whilst the grid concentrates the field and hence modulates the field along the surface. The paramagnetic particles are attracted to the grid surface, and diamagnetic particles (or non-magnetic particles) are moved into the voids of the magnetic grid. The patterned magnetic systems are also produced by etching the permanent magnets.

In many cases, directed movement is needed. This can be realised by polarizing the magnetic particles with a strong homogeneous magnetic field $H_{0}$ that does not exert a force on them. The force is generated by applying an additional small magnetic field h(r) with controlled spatial distribution [30]. The magnetostatic force is proportional to the gradient of h(r):(6)$$F_{m}=\frac{\mu_{0}}{2}\frac{\left|p\right|}{\left|H_{0}\right|}\left(H_{0}\cdot \nabla \right)\;h$$
where p is proportional to $H_{0}$. Attempts were made to design a system to generate h(r) with a linear gradient: $h\left(r\right)=\hat{G}r$, where the tensor $\hat{G}$ is chosen such that (∇·h )=0. A so-called Halbach magnetic system was proposed for this concept [42].

If magnetic particles possess a remanent magnetic moment not parallel to the external field, they experience the action of the magnetic torque T:(7)T=(p×B)

The application of torque to a magnetic particle requires the existence of some degree of anisotropy. Therefore, the particle must exhibit ferromagnetic properties. In contrast, the forces can be imposed in purely paramagnetic or diamagnetic modes.

In the case of an alternating field, the dynamics of p-orientation are found from the torque balance:(8)$$\alpha \frac{d\theta }{dt}=T$$
where α is the rotational friction coefficient, and θ is the angular orientation of the particle (defined by the position of p). For a smooth sphere with a radius b in a fluid with dynamic viscosity η, the parameter $\alpha =8\pi \eta b^{3}$. The values for the torque generated in a typical magnetic tweezer system using micrometre-sized magnetic particles are between 1 and 100 pN μm [43,44]. Magnetic torque is used to mix fluids or to increase the molecular capture rate for biosensor applications.

## 3. Magnetic Field Configurations from Cylindrical Magnets

In the case of a cylindrical geometry and uniform magnetisation in an arbitrary direction, the external magnetic field is calculated analytically. This is very convenient for modelling a required field configuration. The ability to accurately predict the forces in magnetophoretic systems is needed for the optimal design of experimental devices. The two essential geometries include axially and diametrically magnetised cylinders of arbitrary length, as shown in Figure 1.

In general, the static magnetic field is calculated introducing the scalar potential ψ:(9)H=−∇ ψ
which is found using Green’s function technique:(10)$$\psi \left(R\right)=\int_{V}(d{R^{\prime })}^{3}\frac{\sigma (R^{\prime })}{4\pi \left|R-R^{\prime }\right|}\;,\;\;\;\;\;\;\;\;\sigma (R^{\prime })=-𝜵\cdot M$$

In Equation (10), the integration is performed over the cylinder volume, and $\sigma (R^{\prime })$ is the volume magnetic charge density determined by the divergence of the magnetisation M.

For uniform magnetisation (∇·M=0 inside the cylinder), the potential is defined by the surface charge density ($M_{n})$, and integration over volume in Equation (10) is reduced to integration over the cylinder surface.

### 3.1. Cylinder with Axial Magnetisation

In the case of an axially magnetised cylinder (Figure 1a), a convenient method of field calculation is based on representing the cylinder as a set of current loops with the total magnetisation nI (I is the loop current, and n is the number of turns per unit length). In cylindrical coordinates (r,φ,z), the magnetic field can be calculated in terms of generalised complete elliptic integrals [45,46,47]:(11)$$B_{r}=\frac{\mu_{0}aM}{\pi }\left[\alpha_{+}P_{1}\left(k_{+}\right)-\alpha_{-}P_{1}\left(k_{-}\right)\right]$$
(12)$$B_{z}=\frac{\mu_{0}aM}{\pi \left(r+a\right)}\left[\beta_{+}P_{2}\left(k_{+}\right)-\beta_{-}P_{2}\left(k_{-}\right)\right]$$
where $B_{r}$ and$\;B_{z}$ are the radial and axial components of the magnetic flux density, and a is the cylinder radius. The $B_{\varphi -}$ component is absent due to radial symmetry. The parameters $\alpha_{\pm }$, $\beta_{\pm }$, and $k_{\pm }$ involve spatial coordinates and are expressed as
(13)$$\alpha_{\pm }=\frac{1}{\sqrt{{\left(z\pm L\right)}^{2}+{\left(r+a\right)}^{2}}}$$
(14)$$\beta_{\pm }=\frac{z\pm L}{\sqrt{{\left(z\pm L\right)}^{2}+{\left(r+a\right)}^{2}}}$$
(15)$$k_{\pm }^{2}=\frac{{\left(z\pm L\right)}^{2}+{\left(r-a\right)}^{2}}{{\left(z\pm L\right)}^{2}+{\left(r+a\right)}^{2}}$$
where 2L is the cylinder length. In Equations (11) and (12), two auxiliary functions $P_{1},\;P_{2}$ are used:(16)$$P_{1}\left(k\right)=К\left(k\right)-\frac{2}{1-k^{2}}\left(К-Е\right)$$
(17)$$P_{2}\left(k\right)=-\frac{\gamma }{1-\gamma^{2}}\left(𝒫-К\right)-\frac{1}{1-\gamma^{2}}\left(\gamma^{2}𝒫-К\right)$$
(18)$$\gamma =\frac{r-a}{r+a}$$

Equations (16) and (17) involve functions К(k),Е(k), and 𝒫(k), which are calculated using elliptic integrals of the first, second, and third kinds:(19)$$К\left(k\right)=\int_{0}^{\frac{п}{2}}\frac{d\theta }{\sqrt{1-\left(1-k^{2}\right)\sin\theta^{2}}}$$
(20)$$Е\left(k\right)=\int_{0}^{\frac{п}{2}}d\theta \sqrt{1-\left(1-k^{2}\right)\sin\theta^{2\;}}$$
(21)$$𝒫(k)=\int_{0}^{\frac{п}{2}}\begin{array}{c} \frac{d\theta }{\left(1-\left(1-\gamma^{2}\right)\sin\theta^{2}\right)\sqrt{1-\left(1-k^{2}\right)\sin\theta^{2}}} \end{array}$$

The field distribution is shown in Figure 2 with the strongest field gradients around the cylinder edges.

### 3.2. Diametrically Magnetised Cylinder

In the case of diametrical magnetisation (Figure 1b), the method of magnetic potential is used for the field calculations. For this configuration,$\;M_{n}=M\cos\varphi$. For an infinite cylinder, the Laplace equation for the potential ψ in polar coordinates (r, φ) is easily solved:(22)$$\psi =\{\begin{array}{c} \frac{M}{2}r\cos\varphi ,\;\;r<a\\ \frac{M}{2}\frac{a^{2}}{r}\cos\varphi ,\;\;r>a \end{array}$$

For a cylinder of an arbitrary length,
(23)$$\psi \left(r,\varphi ,z\right)=\;\frac{M}{4\pi }\int_{0}^{2\pi }d\varphi^{\prime }\int_{-L}^{L}\frac{a\cos\varphi^{\prime }dz^{\prime }}{\sqrt{r^{2}+a^{2}-2ra\cos\left(\varphi -\varphi^{\prime }\right)+{\left(z-z^{\prime }\right)}^{2}}}\;$$

Equation (23) can be reduced to a form involving elliptical integrals:(24)$$\psi =\frac{Ma\cos\varphi }{\pi }\left[\beta_{+}P_{3}\left(k_{+}\right)-\beta_{-}P_{3}\left(k_{-}\right)\right]$$

Equation (24) is expressed with the help of $P_{3}\left(k\right)$:(25)$$P_{3}\left(k\right)=\frac{1}{1-k^{2}}\left(К-Е\right)-\frac{\gamma^{2}}{1-\gamma^{2}}\left(𝒫-К\right)$$

In the limit of a long cylinder (L≫a), (24) reduces to a simple form, which is very convenient to use:(26)$$\psi =\frac{a^{2}M\cos\theta }{4r}\left(\frac{L-z}{({\left(L-z{)}^{2}+r^{2}\right)}^{\frac{1}{2}}}+\frac{L+z}{({\left(L+z{)}^{2}+r^{2}\right)}^{\frac{1}{2}}}\right)$$

The explicit equations for the field in a general case are:(27)$$H_{r}=-\frac{\partial \psi }{\partial r}=\frac{Ma\cos\varphi }{2\pi r}\left[\beta_{+}P_{4}\left(k_{+}\right)-\beta_{-}P_{4}\left(k_{-}\right)\right]$$
(28)$$H_{\varphi }=-\frac{\partial \psi }{r\partial \varphi }=\frac{Ma\sin\varphi }{\pi r}\left[\beta_{+}P_{3}\left(k_{+}\right)-\beta_{-}P_{3}\left(k_{-}\right)\right]$$
(29)$$H_{z}=--\frac{\partial \psi }{\partial z}=\frac{Ma\cos\varphi }{\pi }\left[\alpha_{+}P_{1}\left(k_{+}\right)-\alpha_{-}P_{1}\left(k_{-}\right)\right]$$

One more auxiliary function entering (27) is of the form:(30)$$P_{4}\left(k\right)=\frac{\gamma }{1-\gamma^{2}}\left(𝒫-К\right)+\frac{\gamma }{1-\gamma^{2}}\left(\gamma^{2}𝒫-К\right)-P_{1}\left(k\right)$$

Figure 3 demonstrates the magnetic induction distribution around a diametrically magnetised cylinder.

We can conclude that the magnetic field generated by a cylinder with arbitrary uniform magnetisation is calculated with the help of elliptical integrals, which can be evaluated using well-defined algorithms, so the computational efforts are minimal. The total magnetic field from an arbitrary number of cylinders is then calculated as a vector sum of fields from each cylinder. This approach allows solving many important problems when designing magnetic systems.

## 4. Basic Magnet Systems for Magnetic Particle Manipulation

### 4.1. Magnetic Particle Manipulation with a Single Magnet

The magnetic properties of cells can be adjusted by the preliminary cultivation of cells with ferromagnetic particles (nanoparticles or magnetic microbeads), which have a much stronger response to a magnetic field. Typically, magnetic nanoparticles (NPs) consist of a small superparamagnetic core 10–20 nm in size within a polymer shell. The NP solution contains clusters (or micelles) of NPs, which are further used to label cells that internalise them and other biological objects. When ferromagnetic particles are cultured with cells in a magnetic field, this causes the aggregation of ferromagnetic particles and magnetomechanical effects on the cell membrane, which affects both the particle uptake and retention by cells [48,49]. Iron oxide NPs are often used for in vivo therapy [19] due to their moderate biocompatibility. However, all NPs are potentially cytotoxic, and their distribution inside the body must be strictly controlled. The cytotoxic effect depends not only on the properties of NPs but also on the cell susceptibility to them. The presence of a static magnetic field can enhance cytotoxicity due to particle aggregates. In the case of iron-oxide NPs, cytotoxicity is also caused by the appearance of reactive oxygen species (ROS), inducing oxidative stress. The application of a gradient magnetic field of moderate strength increases the NP uptake by cells due to additional magnetic forces in the range of tens of pN [22]. Developing programmable cytotoxicity with the injection of iron oxide NPs may provide a therapeutic effect via the induction of oxidative stress in cancer cells [20,21]. This effect was clearly demonstrated by the decrease in the viability of Jurkat cells after treatment with iron oxide NPs in the presence of a magnetic field gradient in the range of 30–40 T/m [23].

A sufficiently strong magnetic field gradient in the order of kT/m is necessary for the directed movement of weak nano- and microsized magnetic objects to overcome the action of various other forces, such as friction, gravity, and thermal forces. Magnets with micron-sized poles are suitable for this purpose. Magnetically soft amorphous microwires [50,51,52] or semi-hard crystalline microwires [53] can be used as these types of magnets. Typical magnetisation curves for soft magnetic microwires with two compositions are given in Figure 4.

The estimated force acting on a paramagnetic particle with a diameter of 1 μm and a susceptibility of 1 (diluted suspension of iron oxide NPs) located at a distance approximately equal to the wire radius is about 2 nN (for the microwire parameters in Figure 4). This exceeds many other forces, as discussed above, but the magnetic force falls off rapidly with increasing distance from the wire.

To evaluate the magnetic particle dynamics in a suspension with insignificant diffusion, the balance between the magnetic force $F_{m}$, Stokes friction force ($F_{S}$) in a viscous medium, and gravity force are considered:(31)$$F_{m}-F_{S}+g\Delta \rho V=0$$
where Δρ is the difference in the density between the particle and the suspension, V is the particle volume, and g is the free-fall acceleration. Considering that the magnetic core occupies only a small volume of the particle, Δρ is small, and the gravity force can be neglected. For a particle of a spherical shape embedded in a medium with dynamic viscosity η, $F_{S}$ is of the form:(32)$$F_{S}=6\pi \eta R_{h}v$$
where $R_{h}$ is the hydrodynamic radius of the particle, and v is the velocity. When the particle is in the vicinity of magnetic poles, it starts to accelerate towards them. Directed movement occurs if the particle finds itself sufficiently close to the poles, where the magnetic force overcomes the action of thermal agitation. This is illustrated in Figure 5 for the system of an axially magnetised microwire and Huh7 cells with internalised iron oxide NPs [28].

Ferromagnetic cylinders magnetised along the diameter can favour the capture of a larger volume of NPs. The magnetic poles are distributed over the wire surface, and the magnitude of the magnetic induction for long wires depends only on the radial coordinate r (for infinite wires, $\left|B\right|=\mu_{0}Ma^{2}/2r^{2}$). Therefore, NPs existing in a cylindrical region around the wire with a diameter equal to several wire diameters will be affected by the field of the wire.

### 4.2. Magnetic Particle Manipulation with Two Axially Polarised Magnets

To control the magnetic field spatial distribution for specific applications, a number of magnets with certain configurations are required. Using two axially magnetised cylinders in the same direction, as shown in Figure 6a,c, it is possible to generate a nearly uniform magnetic field between the poles. Such fields are needed for polarising and clustering magnetic particles. If the cylinders are oppositely magnetised (Figure 6b,d), a zero-field point appears at the centre of the symmetry of the system. An interesting feature is that at more distant areas from this point, the field gradient is almost constant, imposing a constant force on a magnetic particle (Figure 6e). This idea was used to realise magnetophoresis with micelles of iron oxide NPs with a core diameter of about 100 nm [34]. Acrylic well plates with ferrofluid (20 mg/mL micelles) were placed between the same poles of NdFeB magnets with the field distribution shown in Figure 7a. Due to the constant field gradient (Figure 7b), the ferrofluid in all wells moves toward the edges (Figure 7c,d).

A field configuration with a nearly constant spatial gradient can be useful for improving NP penetration in deep tissues. Thus, two cylindrical NdFeB magnets were used to trap stem cells labelled with magnetic NPs at the site of spinal injury [34]. The field created by the two-magnet system can be compared with the concept of two fields: a large constant field and a perturbating field with a linear gradient [30].

### 4.3. Magnetic Particle Manipulation with Two Cylindrical Magnets Magnetised along the Diameter

Long magnets polarised transversally (similar to cylinders magnetised along the diameter, Figure 8a) create quite large areas for capturing magnetic NPs. Pairs of closely spaced cylinders magnetised along the diameter can be a trap for diamagnetic particles [29,54,55]. If the wire diameter is tens of microns, cells with the same low susceptibility as that of water ($-10^{-5}$) can be made to levitate. This is due to the formation of a potential profile with a 2D minimum, including a well-formed camel-like minimum along the wire length, as depicted in Figure 8b,c. The height of the potential barrier near the wire edges increases with increasing length (2L). However, for a longer wire (see Figure 8c), a wide plateau is formed. For the parameters used, the optimal value is L/a≈20. The potential barrier is the highest in the central plane between the wires (x=0).

With this potential landscape, a diamagnetic particle can overcome the gravitational forces. Given that the wire pair is in the horizontal plane and the *y*-axis is the vertical axis, the total energy density is
(33)$$U=-g∆\rho y-\frac{1}{2\mu_{0}}(\chi_{p}-\chi_{ex})B^{2}$$

Figure 9a shows the comparison of magnetic and gravitational forces for a particle density slightly larger than that of water. The second derivative of the total energy with respect to height *y:* $K_{y}=\partial^{2}U/\partial y^{2}\;$is positive in the region with the force balance, demonstrating a stable equilibrium. Figure 9b shows the total energy as a function of (*y*), which has a wide minimum for relatively low values of y/a~1. Therefore, the size of the levitating particle must be sufficiently small compared to the wire radius. The levitation height increases with increasing particle susceptibility; however, this dependence is weak, and increasing $(\chi_{p}-\chi_{ex})\;$by an order of magnitude (from $-10^{-5}\;\mathrm{to}-10^{-4}$) results in an increase in y/a from 0.8 to 1.1. Nevertheless, this demonstrates the possibility of levitation for many types of cells with diamagnetic susceptibility of the order of $-10^{-5}$, which could be increased by the proper contrast. For example, Gd-based contrast agents (CAs) can be used that have paramagnetic susceptibility $\chi_{ex}=1.6\times 10^{-4}$. As Gd-based CAs are currently used in magnetic resonance imaging for medical purposes, their toxicity has already been widely investigated in vivo [56]. However, their cytotoxicity is less studied and should be additionally investigated for different cell lines [57].

### 4.4. Arrays of Magnets

Many applications require arrays of cells to study their behaviour and communication, as they better mimic the cellular processes that occur in individual cells. This simple level of organisation can be used for observational diagnostics and cell therapy and as a starting point for tissue engineering. Many efficient techniques of cell arraying are based on labelling cells with magnetic NPs using immunolabelling or internalisation. Having acquired paramagnetic properties, the cells are attracted to the poles of micromagnet arrays.

The interactions between cancer and normal cells were studied in the 3D microenvironment with the use of micropatterned magnetic structures [58,59,60]. The human non-tumourigenic epithelial cell line MCF10A and its tumour model MCF10A/myr-Akt1, the human metastatic breast cancer cell line MDA-MB-231, and the human dermal fibroblast NHDF were used. For the capture, the cells were labelled with magnetic liposomes and assembled on a pillar-patterned (100 × 100 μm) soft steel plate and permanent magnet (~0.38 T). About 10 cells on each pillar were arranged in the form of a 3D spheroid. In these works, in particular, the effect of fibroblast interaction on the invasive capacity of melanoma was revealed.

In the process of ordering cells in suspension, it becomes difficult to control their spatial distribution without binding. As a contactless trapping technique, diamagnetic levitation can be advantageous [5,57], but it requires strong magnetic field gradients to impose sufficient forces on diamagnetic objects with weak susceptibility (such as cells). Microarrays of strong permanent magnets are needed for this purpose.

There are several ways to produce micromagnetic arrays. The imprinting of hard magnetic powders is micro-scalable and relatively cheap [61]. Thermomagnetic patterning utilises local laser heating of hard magnetic film through a mask [62]. The film is originally uniformly magnetised out-of-plane. During heating, a magnetic field in the opposite direction is applied to cause the remagnetisation of the heated regions. Thus, an array of regions with opposite magnetisation according to the mask design is produced. Permanent magnet alloys can also be deposited onto a micro-patterned silicon wafer, as shown in Figure 10 [57].

Using this micromagnetic system, a diamagnetic trap of Jurkat cells with a low concentration of Gd contrast agent (<10 mM) was realised [57]. The levitation height above the magnetic array plane slightly increased with increasing CA concentration due to an increase in the susceptibility and the relative density. For a CA concentration of 10 mM, the levitation height is estimated to be about 8 microns, which is similar to that obtained with a microwire pair magnetised along the diameter (see Figure 9).

A system comprising a thin film patterned by stripes in the form of a sinusoid was proposed in [63]. Magnetic patterned multilayer thin films of 30 nm iron covered with a 10 nm titanium layer were prepared by photolithography and e-beam evaporation on a glass substrate, followed by a lift-off technique. This patterning resulted in domain wall pinning with a high stray-field gradient at the head-to-head (or tail-to-tail) domain wall regions. In the remnant state after the application of a saturation magnetic field of 0.3 T, a magnetic field gradient of about 2 T/m produced an attractive force of up to 1.3 pN, which was enough to attract cells labelled with magnetic NPs to specific positions. The ability to capture mouse embryonic fibroblasts (FBs) on the film surface was demonstrated.

### 4.5. Methods of Computation of Magnetic Field Produced by Various Magnetic Grids

With the help of analytical equations for calculating magnetic fields of circular cylinders, various micromagnetic arrays can be modelled, as demonstrated in Figure 11.

Figure 12 shows the energy profile represented by $B^{2}$ for a system of closely spaced short cylinders.

In both cases, a deep 2D minimum is formed at the central point of the grid unit cell. For in-plane magnetisation (Figure 12c), the minimum has a wide plateau along the z-direction.

A micromagnetic array with in-plane magnetisation can be made of soft magnetic material as an amorphous ribbon with a typical thickness of 30 μm. The potential wells appear to be sufficient to trap diamagnetic cells with a susceptibility as low as $2\times 10^{-5}$. The modelling results for this case are given in Figure 13.

### 4.6. Microelectromagnetic Patterning

From many issues discussed, it does not matter if the magnetic fields for particle manipulation are generated by permanent or electromagnets. However, with electric magnets, there is a better option to create spatiotemporal fields and realise micropatterning. Attention should be given to the Joule heat release when the dimension of conductors is reduced [64]. We briefly mention some of the achievements in this area.

A two-layer microelectromagnetic matrix of gold planar wires (7 × 7 wires) 10 µm thick and 3 µm in height with a distance between conductors of 20 µm was realised to guide magnetic NPs [65]. Electrical currents of 0.1 A in the top layer and 0.3 А in the bottom layer produced a magnetic field of ~0.4 mT. By changing the current in different parts of the matrix, the maximum field gradient can be created at different nodes, which controllably changes the NP location. In model cells of baker’s yeast (Saccharomyces cerevisiae), the ability to control the location of individual cells and aggregates of several cells was demonstrated [66]. For cell manipulation using a controlled spatial distribution of a magnetic field, a hybrid CMOS/microfluidic device having three-layer microelectromagnets of different shapes was designed [67]. The possibility of moving bovine capillary endothelial cells labelled with 250 nm magnetic beads was demonstrated. A three-layer microelectromagnetic array of planar gold wires (5.5 µm in width and 2 µm in height) was created to capture cells and to study local hyperthermic effects [68]. This device allows collecting important data for further application of magnetic NPs in biology and medicine, for example, for NP certification and the selection of optimal parameters.

## 5. Diffusion of Magnetic Particles

Diffusion is one of the most important processes that control the vital activity of cellular systems. The time scale of intracellular processes that affect many cell functions is highly dependent on the diffusion rate. Thus, adjustment of the diffusion rate can modify cell functions. One possibility is the use of a magnetic field. Here, we do not consider the effect of a uniform magnetic field on electrically charged species [10] or concentration gradients [12].

The dynamic behaviour of small-scale systems is strongly influenced by thermal noise. Thus, the force balance described by Equation (31) involves an additional term, which describes the interaction of a magnetic particle with a dipole moment p with the environment:(34)$$F_{m}=F_{S}+\xi \left(t\right)$$

The last term on the right-hand side is related to energy dissipation due to randomly fluctuating forces caused by thermal noise ξ(t), which obeys the fluctuation–dissipation relation:(35)$$<\xi \left(t\right)>=0,\;\;\;<\;\xi \left(t\right)\xi \left(t^{\prime }\right)>=2\eta k_{B}T\delta \left(t-t^{\prime }\right)$$
where $k_{B}$ is the Boltzmann constant, T is the temperature, $2\eta k_{B}T$ is the noise strength, and δ(t) is the Dirac delta function. A statistical ensemble of magnetic particles is characterised by the probability density c(r,t), which satisfies the combination of the diffusion equation (when $F_{m}=0)$ and the Liouville equation (ξ(t)=0) [69]:(36)$$\frac{\partial }{\partial t}c\left(r,t\right)=\frac{1}{\eta }\nabla \cdot \left(F_{m}\left(r\right)c\left(r,t\right)\right)+D\;\nabla^{2}\left(c\left(r,t\right)\right)$$
where D is the diffusion coefficient.
(37)$$D=\frac{k_{B}T}{\eta }$$

The probability current (or particle flow) is of the form:(38)$$J\left(r,t\right)=-\left(\frac{F_{m}}{\eta }+D\nabla \right)\;c\left(r,t\right)$$
which satisfies the continuity equation:(39)$$\frac{\partial }{\partial t}c\left(r,t\right)+\nabla \cdot \;J=0$$

The diffusion of magnetic particles in the presence of a nonhomogeneous magnetic field resulting in magnetic forces follows Equation (39) with specific boundary conditions, typically corresponding to particle conservation in a restricted area. The change in c(r,t) eventually reaches a stationary distribution, which corresponds to zero flow:(40)J(r,t)=0 

### 5.1. Diffusion of Paramagnetic NPs in a Magnetic Field Produced by Diametrically Magnetised Microwires

Arrays of ferromagnetic microwires with magnetisation along the diameter can be placed inside a suspension with paramagnetic particles to modify their distribution in a relatively large area. In the case of long microwires, the energy $U_{m}\left(r\right)$, which is defined by $B^{2}$ (Equation (22) for the magnetic potential), depends only on polar coordinate r, and diffusion Equation (39) becomes one-dimensional [29]. Figure 14 shows the stationary distribution for different values of particle susceptibility. The stationary concentration around a typical soft magnetic wire (see Figure 3) increases almost twofold, even for weakly paramagnetic particles with a susceptibility of $10^{-4}$. The increase in concentration around the wire proceeds very rapidly, as demonstrated in the inset in Figure 14b. However, this temporal distribution is far from the stationary one.

### 5.2. Regulated Submicron Magnetic Particle Diffusion using Tunable Domain Structure

Stable confinement in a fluidic environment of magnetic particles in 2D can be realised using a tunable magnetic structure. The domain structure in a uniaxial ferrite garnet film is well resolved, as shown in Figure 15a, with a sharp domain wall that can be displaced by a relatively weak magnetic field. The local stray magnetic fields are strong enough to trap and confine microspheres [70]. Moving the domain walls and changing the relative size of domains with opposite magnetisation makes it possible to regulate the diffusive motion of particles [71]. The size of the magnetised domains can be easily manipulated via an external field applied perpendicular to the film plane. Superparamagnetic particles with a relative susceptibility of about 2 are strongly attracted to the magnetic poles. Coating the film with a 1.2 μm thick polymer layer reduces the stray magnetic field but maintains the 2D confinement of particles. Increasing the vertical elevation also modifies the energy profile, as shown in Figure 15b. The energy minima shift towards larger domains. When no field is applied, the potential landscape has small wells, which can be easily overcome by thermal fluctuation. Then, a submicrometric magnetic particle shows diffusive motion without disturbance from the domain structure (Figure 15c). The applied magnetic field increases the domain size with the same magnetisation, which deepens the energy minima separated by stronger potential barriers. This makes it possible to confine the particles between two domain walls (Figure 15d). With this configuration, it was possible to realise a so-called a ‘single file’ system when the particles form a 1D chain.

### 5.3. Enhanced Diffusion of Magnetic Particles Using Travelling Potential (Magnetic Ratchet)

The size of the domains with opposite magnetisation is controlled by an alternating magnetic field with a low frequency, creating a periodically travelling magnetic potential, which is typical for a so-called ratchet system [72]. A characteristic feature of such a system is that the random motion of fluctuating particles can be converted into directed motion. Simultaneously, there are conditions for enhancing particle diffusion.

The magnetic potential $U_{m}$ profile shifts under alternating magnetic field as
(41)$$U_{m}=-U_{0}\cos(k(x-vt),\;\;v=\lambda f,\;k=\frac{2\pi }{\lambda }$$

The parameter $U_{0}$ is defined by the stray field and particle susceptibility. For low frequencies, the particle will follow to the minimum position. However, the average velocity <v> will depend on the applied field frequency. There is a critical frequency $f_{c}$ below which the particle moves with the maximum speed of λf. For $f>f_{c}$, the particle cannot follow the landscape. At high frequencies, the average velocity decreases as $<v>≅\lambda f_{c}^{2}/2f$. Therefore, the thermal fluctuations start to significantly affect the particle speed close to the critical frequency $f_{c}$. The fluctuation effect is typically determined by measuring the mean squared displacement, which can be calculated from particle trajectories. The mean drift across the x-direction (across the domain walls, see Figure 15a,b) is defined as
(42)$$\sigma_{x}^{2}\left(t\right)=<\left(x\left(t\right)-<x(t\right)>{)}^{2}>\sim t,\;<x>=<v>t$$

The effective diffusion coefficient is found from
(43)$$D_{eff}=\underset{t\to \infty }{\lim}\frac{\sigma_{x}^{2}\left(t\right)}{2t}$$

It was demonstrated that the magnetic ratchet can considerably enhance the diffusion coefficient along the propulsion direction in a frequency range close to the critical frequency, as shown in Figure 16a. At low frequencies, the particles are strongly trapped between the potential barriers, and the effective diffusion is smaller. As the frequency approaches $f_{c}$, the potential barriers decrease, enhancing the diffusivity. At higher frequencies well above $f_{c}$, the particles do not interact with the potential landscape, so the diffusion parameter drops to the value observed when no field is applied. This is demonstrated in Figure 16b.

The magnetic ratchet can be realised using soft magnetic microwires with a circular domain structure, as shown in Figure 17. In the presence of a circular domain structure, the stray field is generated at the domain wall location where the magnetisation is directed perpendicular to the wire surface. By applying a very small current of a few mA, the domain wall can be moved, and the effective potential experienced by particles travels along the wire axis.

## 6. Biocompatibility of Ferromagnetic Microwires with a Glass Coating

Individual magnets used as magnetic tweezers can be introduced to a cell culture suspension. In this context, they must be investigated for biocompatibility. Magnetic metal surfaces are usually toxic, so the magnet surface must be protected. In the case of amorphous microwires, they can be produced with a glass coating [51,52], and their biocompatibility and cell survival were investigated in the presence of a strong magnetic field [28,73].

The microwires were placed in the nutrient medium with a cell culture—human embryonic fibroblasts (musculoskeletal tissue). The cell culture was from the collection of the Federal State Budgetary Institution, Federal Research Centre for Epidemiology and Microbiology, named after N.F. Gamaleya. The cells were grown in a nutrient medium—Needle MEM (from the Moscow Institute of Poliomyelitis and Viral Encephalitis)—with 10% foetal calf serum (FCS). The microwires were sterilised by alcohol and ultraviolet light before the experiment. Cell viability was evaluated in two ways: with fluorescent staining by ethidium bromide and MTT: 3-(4,5-dimethylthiazol-2-yl)-2,5-diphenyltetrazolium bromide, also known as thiazolyl blue tetrazolium bromide.

The MTT assay is a calorimetric reaction due to the reduction of MTT to formazan, which is purple and can be measured spectrophotometrically. The investigated cells were seeded in a 12-well container at a concentration of 200,000 cells/mL in each well in a volume of 2 mL of the culture medium with 10% FCS and 0.5 mg/mL MTT and incubated in a CO_2_ thermostat with 5% CO_2_ at 37 °C. Intact cells without microwires grown in parallel with the experimental ones served as controls. The cytotoxicity of microwires was evaluated by measuring the optical density at a wavelength of 545 nm using the Immunochem 2100 photometer. The colour intensity in the wells with the control was taken as a 100% survival rate. No significant difference was detected in the relative optical densities of cell samples with (0.863 ± 0.005) and without (0.848 ± 0.024) microwires, which indicates their non-cytotoxicity.

Cell viability in the presence of microwires was examined visually by fluorescent staining with ethidium bromide, as shown in Figure 18. Cell viability was assessed using a fluorescent microscope, and it was confirmed that the presence of microwires was not toxic to the cell culture. Hence, microwires can be used as sources of gradient magnetic fields for further work with various biological objects.

## 7. Conclusions

This review paper summarises the recent developments in magnetic manipulation used for particle trapping and transport, diffusion, and colloidal assembly. Primarily considered here are systems of permanent magnets since they can be exploited in many applications due to scalability and the absence of heat release. We demonstrate magnetic targeting strategies that have progressed towards clinical trials. However, translation of the technology to clinical practice requires further investigations in the areas of magnetic system design and action mechanisms.

## Figures and Tables

**Figure 1 cells-11-00950-f001:**
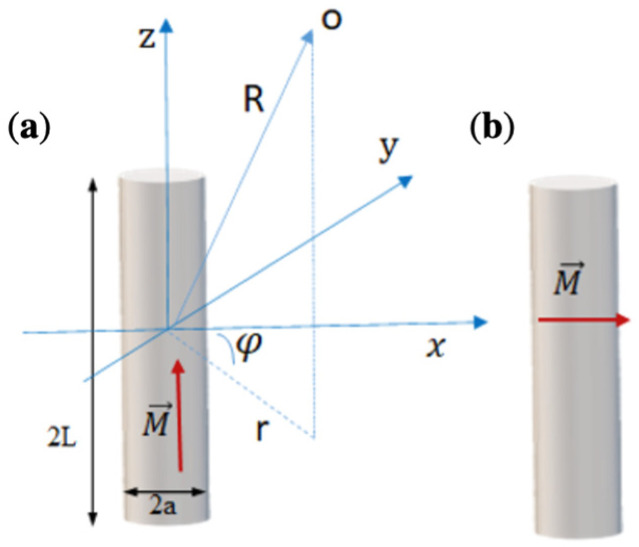
Principle directions and quantities for field calculations of a uniformly magnetised cylinder: (**a**) axial magnetisation; (**b**) diametrical magnetisation.

**Figure 2 cells-11-00950-f002:**
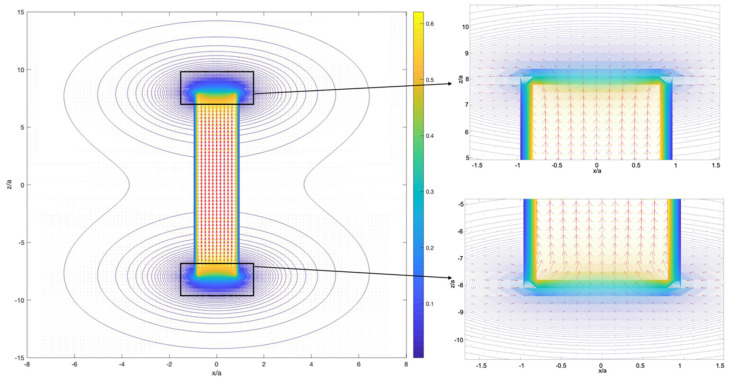
Magnetic induction configuration around a ferromagnetic cylinder with axial magnetisation. The calculations were performed for the following parameters: magnetisation $M=5\cdot 10^{5}$ A/m (Co-based alloys), cylinder radius a=15 μm,  and length 2L=16 a. Colour scale shows the magnitude of |B| in T.

**Figure 3 cells-11-00950-f003:**
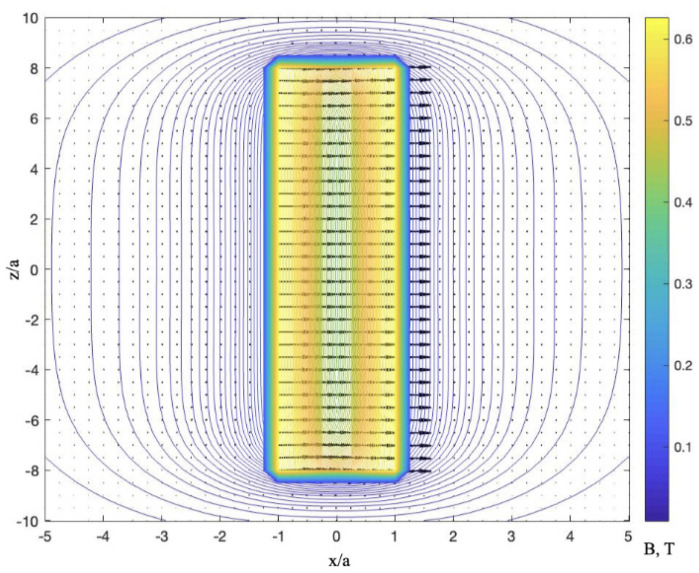
Magnetic induction configuration around ferromagnetic cylinder magnetised along the diameter. The parameters are the same as in Figure 2. Colour scale shows the magnitude of |B| in T.

**Figure 4 cells-11-00950-f004:**
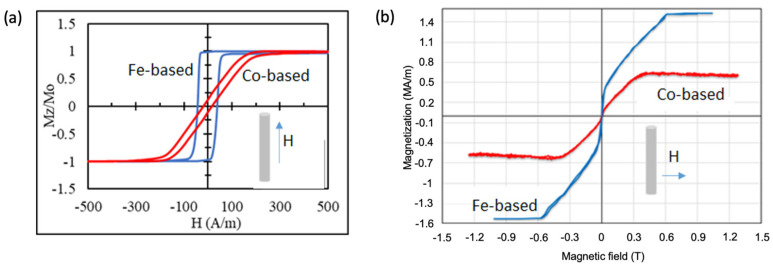
Hysteresis loops of amorphous microwires of Fe-based (Fe_77.5_Si_7.5_B_15_) and Co-based (Co_67.5_Fe_4.5_B_14_Si_11_Cr_3_) compositions magnetised by an axial field (**a**) and a perpendicular field (**b**). The axial and perpendicular curves were measured using an inductive method and vibrating sample magnetometry, respectively. Wires with a diameter of 20–25 microns have a glass coating with a thickness of 4–4.5 microns. The saturation magnetisation is 1.5 MA/m and 0.55 MA/m for Fe-based and Co-based wires, respectively. The wires have different anisotropies: axial for Fe-based and circumferential for Co-based, but both are magnetised along the axis by a small field below 150 A/m. A relatively small field of ~100 mT is required to induce a large perpendicular magnetisation of 0.5 MA/m in Fe-based wire.

**Figure 5 cells-11-00950-f005:**
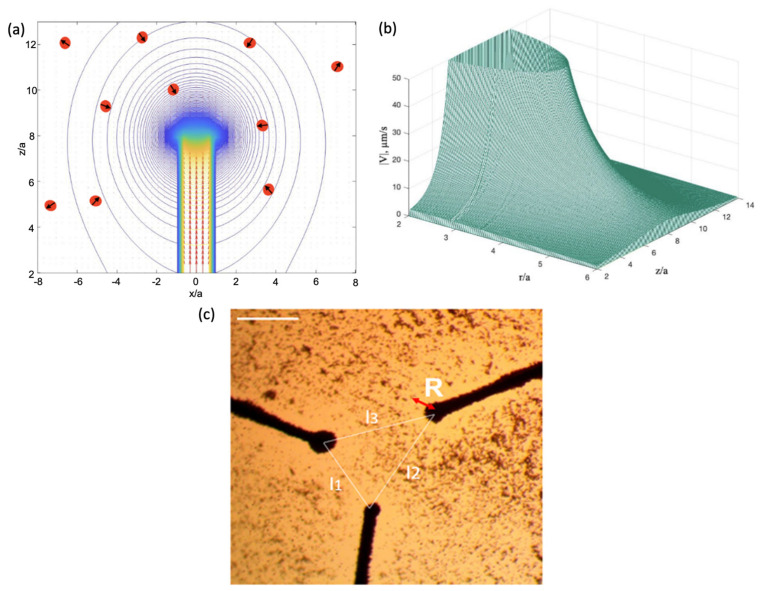
(**a**) Schematic illustration of the paramagnetic particle capture and acceleration towards the wire. (**b**) Particle velocity as it moves towards the wire. For calculations, the field distribution in Figure 2 was used. As particles, cells (human hepatocellular carcinoma line (Huh7)) with internalised iron oxide NPs were considered [28]: $R_{h}=5\;\mu m,$ and the mass of magnetic NPs in the cell is 1 pg with the total volume of $0.2\;{\mu m}^{3}$. Taking $\chi =10^{4}$ for a magnetic nanoparticle, the effective susceptibility of a cell is 2. $\eta =8.9\;\cdot 10^{-4}$ Pa × s (viscosity of water). (**c**) Distribution of iron oxide NPs around amorphous microwires of Co-based alloy with a diameter of 30 microns after the application of a small magnetic field that magnetises the wires along the axis. Areas around microwire edges are free from particles. Arrow *R* designates the size of particle-free area. Scale bar equals 200 µm. (**c**) is reprinted with permission from ref. [28]. Copyright 2020, Elsevier B.V.

**Figure 6 cells-11-00950-f006:**
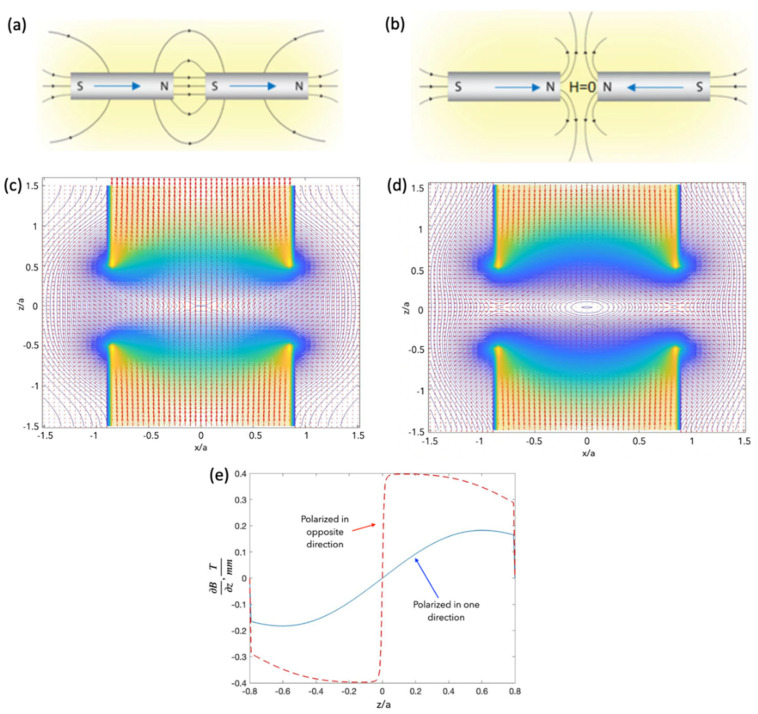
Schematics of magnetic field configuration with two axially magnetised cylinders: (**a**,**c**) cylinders with the same magnetisation; (**b**,**d**) oppositely magnetised cylinders. Closely spaced magnets with the same polarisation produce a nearly uniform magnetic field between N and S poles, whilst the magnets with opposite polarisation have a zero-field point (between alike poles) in the centre of the magnet system. (**e**) Gradient of magnetic induction. Calculations in (**c**–**e**) were performed for typical permanent magnets (NdFeB magnets with the surface field of 0.66 T).

**Figure 7 cells-11-00950-f007:**
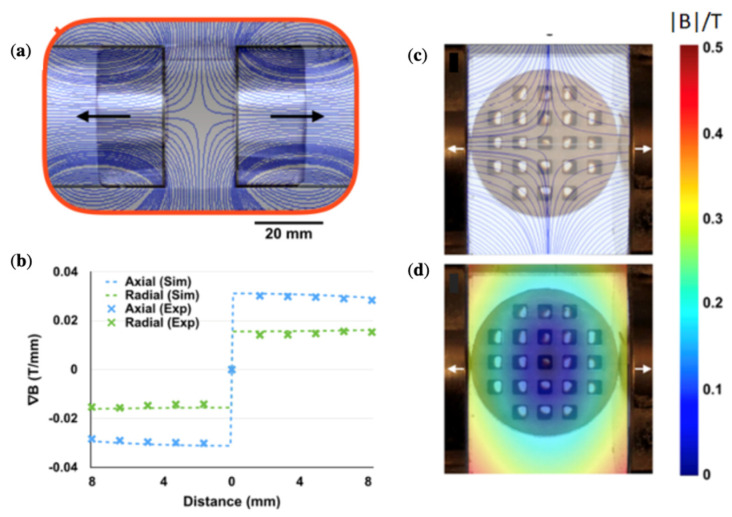
Magnetic field (**a**) and field gradient (**b**) distributions for two NdFeB magnets with opposite polarisation (magnet dimensions: 25 mm in diameter and height, 20 mm between the magnets). The surface field is 0.66 T. The magnetic device contains a sharp zero point surrounded by constant field gradients. (**c**,**d**) Images of the ferrofluid redistribution in acrylic wells placed between the magnet poles. Reprinted with permission from ref. [34]. Copyright 2020, American Chemical Society.

**Figure 8 cells-11-00950-f008:**
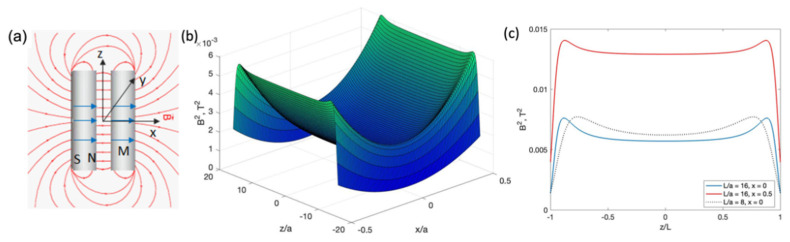
(**a**) Magnetic field configuration from a pair of cylindrical magnets polarised along the diameter. (**b**) Two-dimensional energy plot represented by the square of the magnetic field induction ($B^{2}$) from a pair of microwires in the *x*-*z* plane. y/a=1. The magnet parameters correspond to Co-based amorphous microwire (see Figure 4) with radius a=10 μm,  length 2L=32a, magnetisation M=0.5 MA/m, and distance between wires d=3a. (**c**) Distribution of $B^{2}$ along the wire length (*z*-axis) for two different lengths, L/a=8 and 16. x=0 and 0.5/a, y/a=1. A saddle-shaped minimum with a wide plateau is formed between the wire edges, and the energy peaks near the edges are sharper with increasing length. Moving away from the center point on the x-axis reduces these peaks.

**Figure 9 cells-11-00950-f009:**
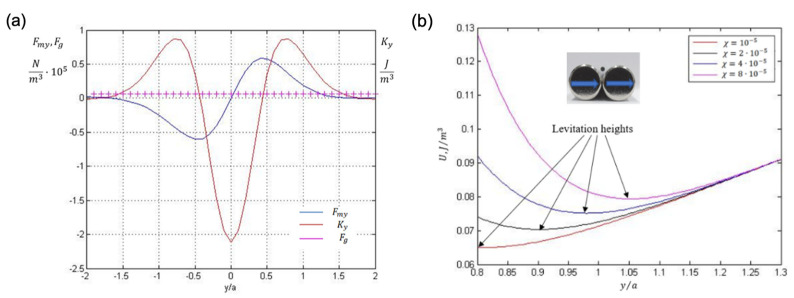
(**a**) Dependence of the *y*-component of magnetic force $F_{my}$ and the second derivative of the total energy $K_{y}=\partial^{2}U/\partial y^{2}\;$ on height y above the pair in the symmetry point(x=0, z=0). Gravitational force density $F_{g}=g∆\rho =0.2\times 10^{4}\;N/m^{3}$ (with respect to suspension), $(\chi_{p}-\chi_{ex})=-10^{-4}$. (**b**) Total energy density vs. the distance (*y*) for different values of the diamagnetic susceptibility. The position of levitation is indicated by arrows. The wire parameters correspond to Figure 8. Inset shows the levitation configuration.

**Figure 10 cells-11-00950-f010:**
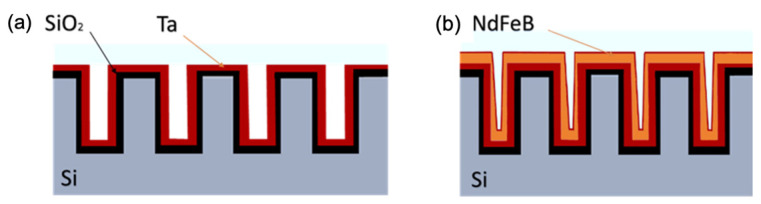
Schematics of microfabrication process to produce grids of micromagnetic poles with NdFeB. (**a**) Silicon wafer is micropatterned using lithography and etching and coated with SiO_2_ layer 100 nm thick, and then 100 nm thick Ta layer is sputtered to serve as a reaction barrier between SiO_2_ and NdFeB; (**b**) 30 μm thick NdFeB layer is sputtered and coated with 500 nm Ta layer.

**Figure 11 cells-11-00950-f011:**
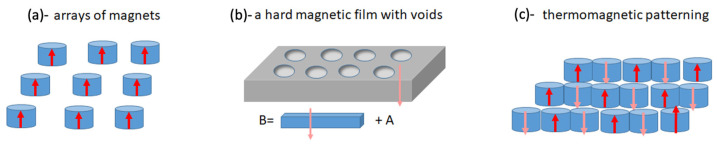
Schematics of micromagnetic arrays. (**a**) Basic system of uniformly magnetised cylinders (along the height or along the diameter); (**b**) uniformly magnetised magnetic film with voids that magnetically is equivalent to the combination of a uniform film and a system of cylinders magnetised oppositely to the film; (**c**) an array produced by thermomagnetic patterning corresponds to an array of closely spaced cylinders with mutually opposite magnetisation.

**Figure 12 cells-11-00950-f012:**
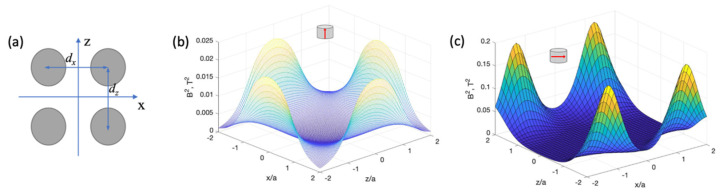
(**a**) Schematic configuration of a unit cell of arrays of cylindrical micromagnets. (**b**,**c**) Two-dimensional plots of the energy profile (per unit cell) represented by the square of the magnetic field induction ($B^{2}$) from arrays of cylindrical micromagnets with axial and diametric magnetisations, respectively, in the x-z plane. *y*/*a* = 0.7. The magnet parameters correspond to Co-based amorphous alloys with magnetisation $\mu_{0}M=0.6\;T$, radius *a* = 15 μm, length 2*L* = 2*a*, and distance between magnets *d*_x_ = *d*_z_ = 2.5*a*.

**Figure 13 cells-11-00950-f013:**
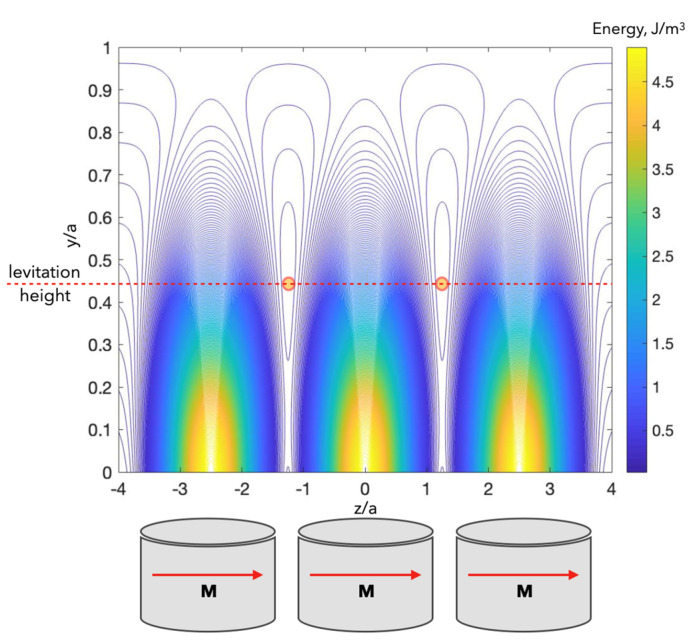
Equipotential energy density U (*x* = 0, *y*, *z*) for a periodic array of cylindrical magnets with in-plane magnetisation (Figure 12a,c). The total energy U includes magnetic and gravity energies (Equation (33)). $g∆\rho =0.2\times 10^{4}\;N/m^{3}$; $(\chi_{p}-\chi_{ex})=-0.5\times 10^{-4}$. The change in the z-coordinate is shown along three micromagnets (depicted in the figure), and the graphical image can be periodically continued along the z-axis.

**Figure 14 cells-11-00950-f014:**
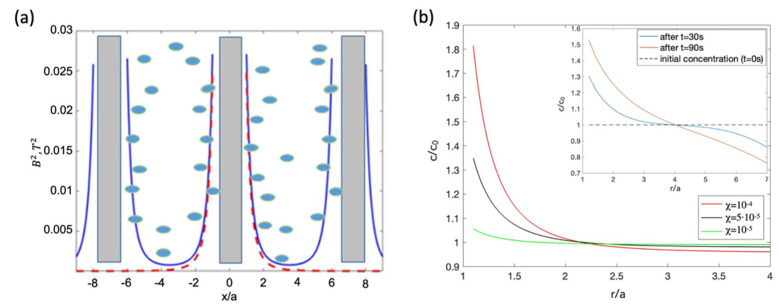
(**a**) Magnetic energy profile from an array of microwires magnetised along diameter (blue curves; red curves show the energy profile from a single wire) and schematics of paramagnetic NP distribution with a higher concentration near the wires. (**b**) Stationary distribution of concentration $c/c_{0}$ (normalised to initial concentration ${\;c}_{0}$) around a single microwire for different values of magnetic susceptibility. Inset shows the concentration distribution after a few characteristic times. r/a is the normalised polar coordinate. The calculations were performed for the parameters:$\;D=3\times 10^{-12}m^{2}/s$, $\chi =10^{-4}\;\left(for\;insert\right),$ $V=10^{-21}{\;m}^{3}$, a=15 μm, and M=0.5 MA/m.

**Figure 15 cells-11-00950-f015:**
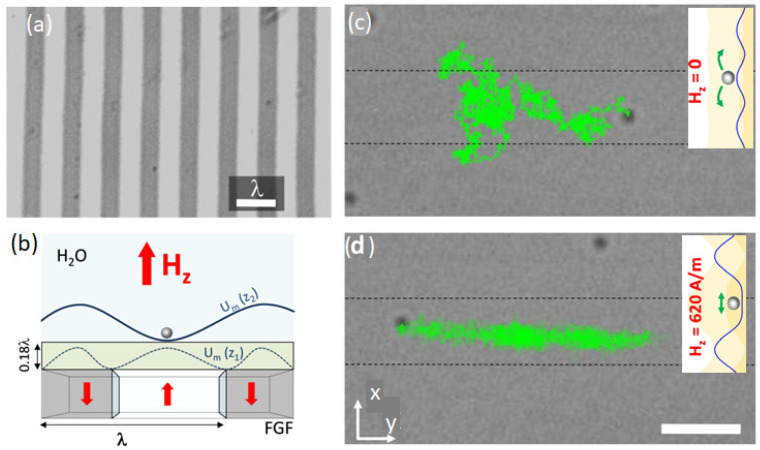
(**a**) Image of a uniaxial ferrite garnet film with magnetic domains having a spatial periodicity λ = 6.8 µm, visualised due to the polar Faraday effect. (**b**) Domain configuration under applied field. The lines above demonstrate the change in the potential landscape with the distance above the film. (**c**,**d**) Microscope snapshots of the trajectories of one particle with a diameter of 360 nm without the field and in the presence of the field, respectively. The dashed lines designate the position of the domain walls. Insets show the potential landscapes along *y*-direction. The scale bar is 5 µm. Reprinted with permission from ref. [71]. Copyright 2016, American Chemical Society.

**Figure 16 cells-11-00950-f016:**
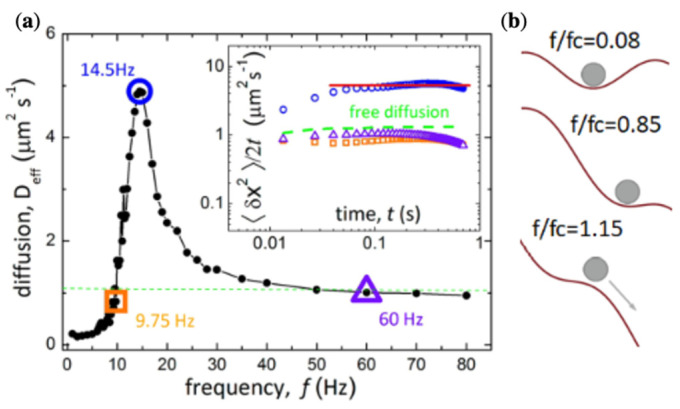
(**a**) Effective coefficient of diffusion $D_{eff}$ measured across the domain walls of the ferrite garnet film (presented in Figure 15) as a function of the external field frequency for superparamagnetic particles with a diameter of 270 nm and susceptibility of 2. The field amplitude is 1200 A/m. The green dashed line represents the diffusion coefficient measured at a zero field. The frequency of the diffusion maximum $f_{m}$ = 14.5 Hz is close to the critical frequency $f_{c}$ = 13.4 Hz. The inset shows $\sigma_{x}^{2}\left(t\right)/2t$ for different frequencies marked in the main plot. The red line for $f_{m}$ is used to determine $D_{eff}$. (**b**) Schematic representation of the magnetic potential landscape for different frequencies. Reprinted with permission from ref. [72]. Copyright 2016, American Chemical Society.

**Figure 17 cells-11-00950-f017:**
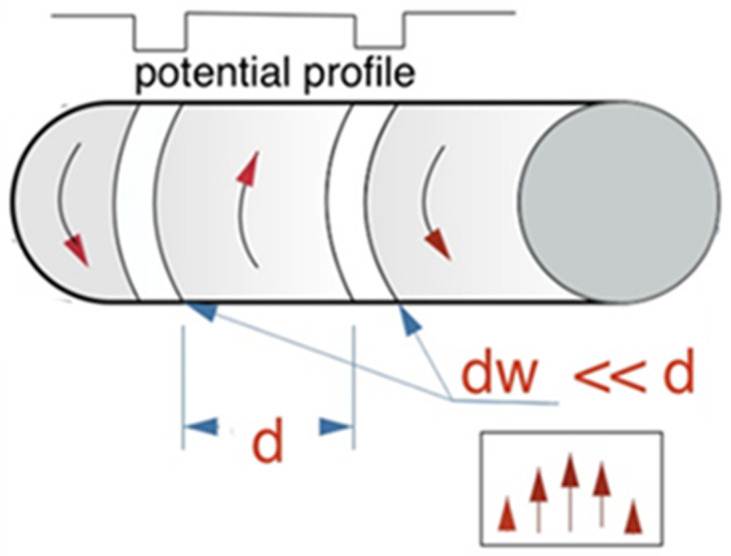
Realising a travelling magnetic potential in a microwire with a circular domain structure by passing an alternating current through the wire.

**Figure 18 cells-11-00950-f018:**
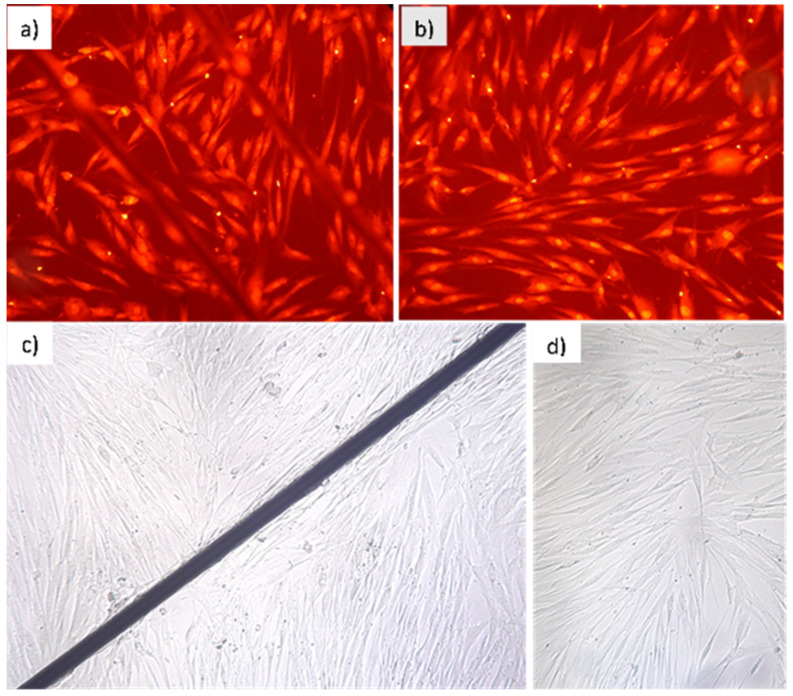
Assessment of cell viability in the presence of microwires (**a**,**c**) in comparison with control samples without the wires (**b**,**d**). A nutrient medium with the cell culture—human embryonic fibroblasts (FEH-T)—was used. (**a**,**b**) After exposure for 24 h, fluorescent staining by ethidium bromide was used to assess the survival of cells in the presence and in the absence of microwires. Luminosity was examined under a fluorescent microscope. (**c**,**d**) Optical microscope images after 168-h exposure demonstrate no visual effect of microwires on cell viability.

## Data Availability

Not applicable.

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
