# Peer review of "Spatial Manipulation of Particles and Cells at Micro- and Nanoscale via Magnetic Forces"

_cells, 2022, doi:10.3390/cells11060950_

Round 1

Reviewer 1 Report

In recent years, magnetic fields have been increasingly used in biomedicine. In fact, therapies have emerged that use the principle of magnetic forces. This review gives us an overview of the application of these forces in biomedical technology, taking into account its benefits and limitations. In my opinion, it is much more written and very explanatory, being complemented with very good illustrations. In addition, english should be revised by a native English-speaking.

Author Response

English was revised with the help of English-language specialist through out the manuscript. The changes are marked in yellow.

Reviewer 2 Report

This paper entitled "Spatal manipulation of particles and cells at micro and nanoscale via magnetic forces" reported the recent developments in magnetic manipulation used for particle trapping and transport, diffusion and colloidal assembly. In this paper, various magnetic systems capable of generating magnetic fields with required spatial gradients are analyzed. Starting from simple systems of individual magnets and methods of field computation, more advanced magnetic microarrays obtained by lithography patterning of permanent magnets are introduced. More flexible field configurations can be formed with the use of soft magnetic materials magnetized by an external field which allows control over both temporal and spatial field distribution. This work is significant and very interesting. However, there are still some problems in the article that need to be corrected.

(1) A minor issue, there should be a space between the number and the unit. For example, line 293.

(2) The overall resolution of the pictures in the manuscript is not enough, and the clarity needs to be improved. The layout of the pictures in the text should be adjusted appropriately to make it more tidy and beautiful.

(3) Suggested list of recent literature on magnetic field regulating interactions between particles and cells. The mechanism by which the magnetic field manipulates particle-cell interactions should be elucidated in detail.

(4) In Figure 8c, L/a appears to be 16 in the legend, but 10/20 in the title, which seems contradictory.

(5) In Figure 18, photos processed at different times should be displayed uniformly.

(6) The location of Figure 18 in the text has been lost.

(7) The format of references should be uniform.

(8) The authors discuss extensively how gradient magnetic fields and forces are calculated, but how much force can drive paramagnetic or diamagnetic particles to move in a magnetic field.

Author Response

  • A minor issue, there should be a space between the number and the unit. For example, line 293.

We have carefully proof-read the manuscript and corrected these and other typos.

  • The overall resolution of the pictures in the manuscript is not enough, and the clarity needs to be improved. The layout of the pictures in the text should be adjusted appropriately to make it more tidy and beautiful.

We have improved majority of figures: 4b, 5,6,8,9,11,12,14 increasing resolution, enlarging the legends.    Figure 18 was completely changed. The layout of the figures was also improved.

  • Suggested list of recent literature on magnetic field regulating interactions between particles and cells. The mechanism by which the magnetic field manipulates particle-cell interactions should be elucidated in detail.

This issue was discussed in relevant places (marked with grey). In short, magnetic properties of para- or diamagnetic cells can be adjusted by preliminary cultivation of cells with ferromagnetic particles (nanoparticles or magnetic microbeads), which have a much stronger response to a magnetic field. When ferromagnetic particles are cultured with cells under a magnetic field, it causes aggregation of ferromagnetic particles and magnetomechanical effects on the cell membrane, both affecting particle uptake and retention by cells. A gradient magnetic field also increases the particle uptake due to magnetic forces. Additional references were added (current references 22, 43-44).

  • In Figure 8c, L/a appears to be 16 in the legend, but 10/20 in the title, which seems contradictory.

This was corrected accordingly. Two values of L/a were considered in Fig. 8c: 8 and 16.

  • In Figure 18, photos processed at different times should be displayed uniformly.

This figure was changed in response to the comments of the academic editor.  We placed fluorescent images after 24 hours (with wires and without wires) and optical images after 168 hours (also with wires and without wires).

  • The location of Figure 18 in the text has been lost.

Corrected.

  • The format of references should be uniform.

All the references are now  given in the same style according to the journal requirements.

  • The authors discuss extensively how gradient magnetic fields and forces are calculated, but how much force can drive paramagnetic or diamagnetic particles to move in a magnetic field.

We have added more comparative estimates of magnetic force  for various conditions. In general, the  force of the order of 10 pN is sufficient to overcome the viscous force of a microsized particle in the liquid and, therefore, to transport it. Additional references are also given (current references 16, 40, 41).